# Reconstruction and Efficient Visualization of Heterogeneous 3D City Models

**Mehmet Buyukdemircioglu** and **Sultan Kocaman** *

Department of Geomatics Engineering, Hacettepe University, Ankara 06800, Turkey;
mbuyukdemircioglu@hacettepe.edu.tr
* Correspondence: sultankocaman@hacettepe.edu.tr

**Abstract:** The increasing efforts in developing smart city concepts are often coupled with three-dimensional (3D) modeling of envisioned designs. Such conceptual designs and planning are multi-disciplinary in their nature. Realistic implementations must include existing urban structures for proper planning. The development of a participatory planning and presentation platform has several challenges from scene reconstruction to high-performance visualization, while keeping the fidelity of the designs. This study proposes a framework for the integrated representation of existing urban structures in CityGML LoD2 combined with a future city model in LoD3. The study area is located in Sahinbey Municipality, Gaziantep, Turkey. Existing city parts and the terrain were reconstructed using high-resolution aerial images, and the future city was designed in a CAD (computer-aided design) environment with a high level of detail. The models were integrated through a high-resolution digital terrain model. Various 3D modeling approaches together with model textures and semantic data were implemented and compared. A number of performance tuning methods for efficient representation and visualization were also investigated. The study shows that, although the object diversity and the level of detail in the city models increase, automatic reconstruction, dynamic updating, and high-performance web-based visualization of the models remain challenging.

**Keywords:** 3D urban scene modeling; photogrammetric reconstruction; digital terrain models; data fusion; visualization; efficient representation; texturing; CesiumJS; virtual reality

## 1. Introduction

The World Urbanization Prospects describe the future of population in urban areas, whereby more than half of the world population is living in cities [1]. The urban sprawl (i.e., 30% of the world's population in 1950 vs. 55% in 2018) brought several challenges to life in cities, such as poor accessibility due to insufficient transportation infrastructure, poor air quality, inequality, safety, and slums, which are unplanned settlements, as described by the United Nations Sustainable Development Goal 11 (Sustainable Cities and Communities) [2]. The efforts in tackling the problems that cause the reduction in life quality are increasingly discussed under the smart city concept, which provides alternative solutions to the traditional urban planning approaches using new technologies [3–5]. On the other hand, the smart cities are seen as technology-driven entities, while they also need to ensure the inclusion of citizens and the political, social, economic, and knowledge stakeholder groups in solving problems [5].

The three-dimensional (3D) GIS (geographical information system) with 3D models of urban structures (e.g., buildings, city furniture, plans, vegetation, infrastructure, etc.) provides the technological basis for the design and development of smart cities, which require site-specific design. Urban simulation and visualization are useful for a variety of applications, such as regional planning, transportation, land-use regulations, and environmental protection [6]. The digital transformation

of 3D spatial data and their standardized data models bring advantages in digital space and makes the decision-making process more illustrative, easier to understand, and more comprehensible [7]. The efforts toward the realization of 3D GIS involve accurate and detailed 3D modeling of objects with semantic attributes and high-performance visualization. Models coming from various sources can be integrated using widely accepted exchange formats, such as City Geography Markup Language (CityGML) and City JavaScript Object Notation (CityJSON). CityJSON [8] is an efficient alternative to CityGML avoiding the use of GML, which can be tricky for implementation. The latest version of CityGML (i.e., 3.0) [9] is expected to introduce various additional features for the management of multiple versions and time-dependent properties. The exchange formats allow generating the models with semantic data.

A 3D city model represents buildings and other man-made and natural objects in an urban area, together with semantic information and a digital terrain model (DTM), which also reflects the relations between them [10]. The 3D city models usually consist of a DTM and buildings, as well as street-space and green-space models [11]. Although other city objects such as trees, streets, tunnels, etc. may increase the sense of realism, they are usually represented in a 3D model when they have a specific role in the desired application or analysis, especially for visualization. The quality of 3D city models is determined by geometric, topological, and semantic accuracy [12]. On the other hand, they require more storage space in comparison to two-dimensional (2D) data [13], and integration of data from various sources is required to create model geometries with texture and semantic information. The semantic and topological information added to the polygonal components of buildings, such as year of reconstruction, energy consumption, walls of a building, etc., can be stored in a database and linked with the building object.

The 3D city models are also crucial for planning in a virtual environment for the management of cities, for performing different planning simulations, risk and disaster scenarios, etc. The third dimension eliminates the disadvantages of 2D when considering multi-level structures [14]. The 3D geometry allows the users to have a more natural interaction with the geo-spatial data [15]. Association of geographical and semantic data in 3D city scenes makes them easily understandable for stakeholders, decision-makers, citizens, and machines. The efforts carried out by both public and private organizations in generating 3D city models are increasing, and many impressive and realistic 3D city models are regularly offered at scientific, professional, and commercial events (e.g., References [7,16–18]).

A smart city concept development project was initiated by the Ministry of Environment and Urbanization (MoEU), Turkey, in 2018 [15] with the aim of designing a future city district having green, human-centric, smart, and safe qualities, along with a sense of place, by reflecting the local expectations in an unconstructed area. For this purpose, an area of 287 hectares was declared as a project site in Şahinbey District of Gaziantep City, Turkey, by the MoEU. As a part of the project, a 3D GIS environment was established by modeling the neighboring settlement areas of the project site using aerial photogrammetric datasets and combining them with the design elements created in different CAD (computer-aided design) environments by urban planners, landscape architects, environmental and computer scientists, and geological engineers.

In this study, the model reconstruction and the visualization methodology of the integrated city model is presented and analyzed. The main aim of the study was to investigate different modeling and visualization options for the city model obtained from multiple sources (i.e., photogrammetric datasets and CAD elements), to assess their performance and to report various issues encountered during the implementation of the 3D GIS environment. Examples of such efforts in integrating existing models with user-generated model content are increasing in the recent literature (e.g., References [19,20]). Here, the future city contains a high level of detail (LoD) defined according to the CityGML standard [21], LoD3, whereas the existing districts were reconstructed as textured LoD2 models using photogrammetric techniques. The integrated model was firstly stored in CityGML format, and it was visualized together with a true orthophoto basemap and a high-resolution DTM. Open-source CesiumJS virtual globe and Unity game engine with virtual reality (VR) support were implemented as visualization

platforms and evaluated for different aspects. The generated models were stored in CityGML format to ensure interoperability between different applications and semantic data integration. For efficient and high-performance visualization, various optimization methods were applied to the generated models. Additional web-based functions such as measurement tools, as well as querying and styling of the generated model, were analyzed in detail. In addition, the advantages of model exploration in the VR environment were pointed out.

This article is structured as follows: the introduction (Section 1) is followed by literature examples of 3D city model representations and different use cases (Section 2). In Section 3, the study area characteristics, overall methodological workflow, and the datasets are presented. Model implementation results of the existing LoD2 city model and future smart city concept in LoD3, as well as encountered problems and their solutions, are explained in detail in Section 4. Further details on the virtual web globe and Unity game engine visualization methodology and the results are explained in Section 5. Section 6 is dedicated to the discussion. Conclusions drawn from the study are provided in the final section with future recommendations.

## 2. Related Work

Although the development of 3D GIS was initially limited to urban planning and visualization [22], with the recent developments in computer software, hardware, and new reconstruction methods, it expanded beyond these limits. With recent applications such as locations, conditions, trends, patterns, and models, 3D GIS platforms are becoming more practical [23]. For example, 3D city models were applied in different use cases in the literature, such as noise mapping [24], web-based visualization [25], volunteered geographic information [26], energy demand estimation [27], forest simulation [28], augmented reality [29], real estate [30], air quality [31], tsunami analysis [32], protected area planning [33], cultural heritage applications [34], campus models [35], etc.

The city models can be reconstructed manually [36], semi-automatically [37], or automatically [38] from different data sources such as satellite imagery [39,40], large-format nadir aerial imagery [37], oblique aerial imagery [41], unmanned aerial vehicle (UAV) imagery [42], LiDAR (light detection and ranging or laser imaging detection and ranging) point clouds [43], mobile laser scanning data [44], or joint processing of different data sources [45–48]. Procedural city/building modeling is another approach for producing extensive architectural models for computer games and movies, as well as smart city concepts with high geometric detail at low cost, which are currently integrated in the CityEngine Tool of ESRI (Redlands, CA, USA) [49,50]. With the latest developments in machine learning methods and computer hardware, 3D city models can also be reconstructed using deep learning approaches [51–53].

Three-dimensional visualization is defined as a field that provides tools for exploration and analysis of spatial data [54], and it is popular in various disciplines [55,56]. A comprehensive review showed that the majority of analyzed use cases had visualization as the main goal [55], which demonstrates that visualization is an important focus of such models [56]. With the recent developments in web and Internet technologies, access to services and spatial information from private individuals or business applications is faster and easier than before [57]. The cartographic rules that can be applied to web-based 3D city models are neither standardized nor fully investigated so far [58], although a number of applications can be found in the literature (e.g., References [59,60]). Herman and Řezník [61] compared different web technologies and data formats used in 3D model visualization in their study. The recent developments in web technologies such as WebGL (Web Graphics Library) [62] and HTML5 (Hypertext Markup Language) led to new web-based visualization environments and products such as CesiumJS [63], three.js [64], and Unity [65]. With WebGL technology, users can integrate and visualize 3D content on web browsers without any need for a plugin or extension. Furthermore, 3D city models can be visualized on different platforms such as virtual web globes, smartphones [66], and game engines for exploring them with VR [67,68]. The low cost and the widespread use of mixed reality

and VR technologies allow urban simulations to be explored in game engines, such as Unity, Unreal Engine, or CryEngine, which also offer free access to developers.

Virtual globes are now a standard way of visualizing and analyzing different types of geospatial data for different disciplines [69–72]. Google Earth [73], Nasa World Wind [74], iTowns [75], and CesiumJS are examples of virtual web globes. However, the use of 3D models on the web has some technical challenges such as network speed and cross-platform support, and, due to the rare support of 3D graphic application programming interface (API) cross-platform migration, developing a 3D GIS environment that works on different operating systems and devices such as Windows, Linux, Android, or iOS is costly and requires more effort [76].

Interoperability of 3D city models still remains as a challenging task and still has many issues to overcome [19,20,77]. CityGML [78] is an XML-based open-source data format for storing and managing semantic 3D city models, and it is adopted by Open Geospatial Consortium [79] as an international standard format. Although it supports interoperability between different applications, it is not an appropriate data format for visualizing city models directly on a web platform [80]. For efficient visualization, CityGML data need to be converted into another format like COLLADA (Collaborative Design Activity) [81], Extensible 3D (X3D) [82], or 3D Tiles [83]. Murshed et al. [84] found that 3D Tiles is a suitable alternative for visualizing big volumes of 3D city models due to its tiling property. Chaturverdi et al. [85] developed an interactive web client based for semantic 3D city models using HTML5 and WebGL. Wendel et al. [58] developed a plugin-free web-based visualization semantic 3D city model of Karlsruhe, Germany. Farkas [86] compared the most capable open-source web GIS libraries based on their feature coverage. Chen et al. [87] developed a workflow for visualizing BIM (building information modeling) models on CesiumJS. Resch et al. [88] developed a 3D web-based interface for visualizing marine data and pointed out the problems for visualization of four-dimensional (4D) data (time as the fourth dimension). Four-dimensional data visualization can increase the efficiency of exploring relationships in complex geospatial data with large volumes [89]. Three-dimensional city models are still mostly static, and adding the fourth dimension dynamically on the on web is an emerging research area.

Three-dimensional terrain visualization also plays an important part in 3D city models, urban visualization, and many more GIS applications [90]. The main reasons for visualizing the terrain along with the 3D city model are (a) provision of a more realistic globe to the stakeholders, (b) completeness of earth surface data, (c) higher reliability in decision-making process, and (d) suitability for a wider range of applications such as engineering, disaster management, planning, etc. Visualizing terrains with high volumes requires advanced rendering strategies [91] and efficient level of detail approaches [92]. Although several studies exist on the high-performance rendering of high-resolution terrains, the topic still remains challenging. There are a few strategies in the literature for reducing the data volume for the terrain and other geodata [93]. Campos et al. [94] explained the level of detail concept for terrain models based on visible details depending on the perception of the user at a given a point of view, and they tackled the problem of generating world-scale multi-resolution triangulated irregular networks optimized for web-based visualization. Staso et al. [95] developed a multi-resolution and multi-source terrain builder for CesiumJS.

Although there are several studies on different components or stages of 3D city model generation and visualization, as briefly summarized above, further researches and investigations in this field are required due to the complexity of the problem. Modeling our environment in 3D by adding the time and semantic information is essential for many applications, and comprehensive investigations with real-world data can facilitate the developments.

## 3. Materials and Methods

### 3.1. Study Area

Gaziantep is located at the intersection of the Mediterranean and Southeastern Anatolia Region of Turkey and is one of the oldest cities, inhabited for thousands of years. It is the most urbanized city of the Southeastern Anatolia Region with a population over two million, developed industrial units, and genuine culture. Şahinbey District located in Gaziantep City is Turkey's third most populated municipality with a population size of nearly one million. It has an area of 960 km² and a mean altitude of 850 m above sea level. The 3D model of a part of Şahinbey was generated in this study and merged with the smart city design models by (a) reconstructing the 3D model of the existing neighborhoods using aerial photos obtained from a photogrammetric flight mission, and (b) conversion and georeferencing of the solid 3D model of the designed smart city in the pilot project. Both models were integrated in a joint reference system and combined as a single city model. A general overview of the study area, the boundaries of the modeled city parts and the designed future city, and the perspective center positions of the aerial images used in the study are given in Figure 1.

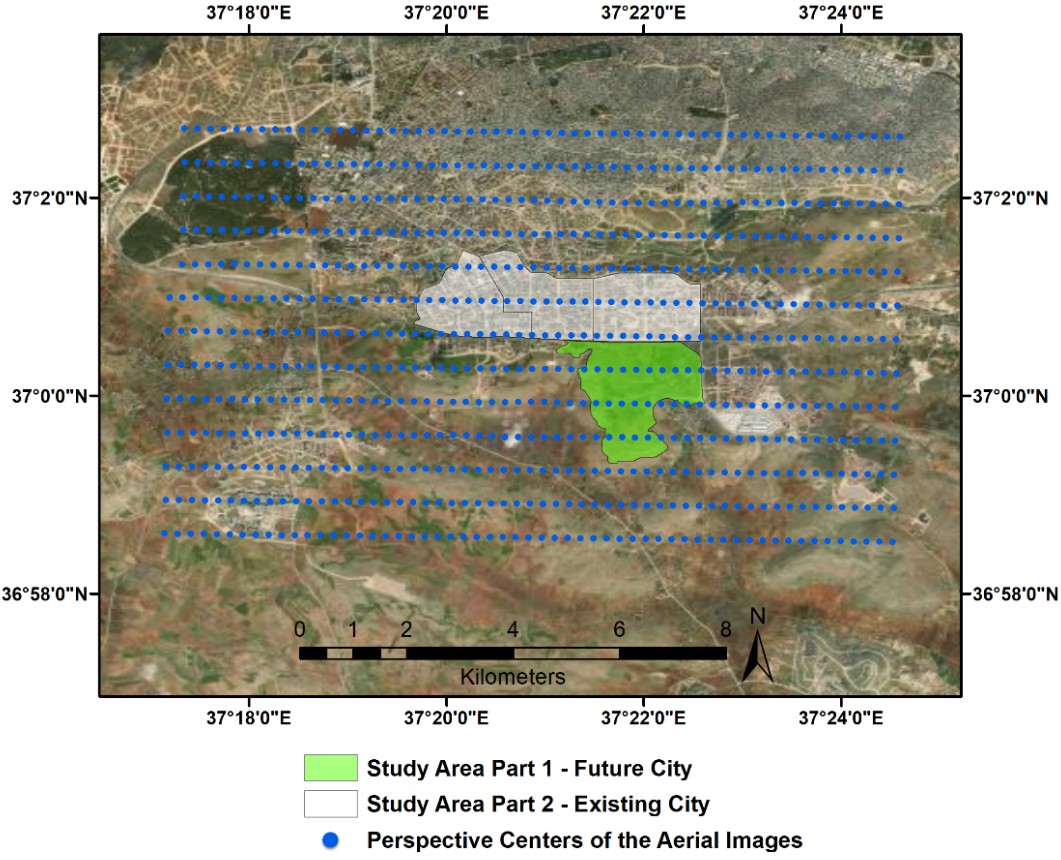

**Figure 1.** A general view of the study area in Sahinbey, Gaziantep with the boundaries of the reconstructed three-dimensional (3D) model of existing city parts (white) and the designed future city (green). The perspective center positions of the aerial photos used for the reconstruction are depicted with the blue points.

### 3.2. Overview of the Methodology

The study workflow consists of three main stages as shown in Figure 2. The first stage is the generation of the LoD2 model of existing neighborhoods of the project area using aerial photogrammetric datasets. This model was also employed during the design stages of the future city model for visualization purposes. The second stage was the conversion of the LoD3 building models designed

by the architects. Since the architectural designs were not geolocated and were not suitable for LoD3 model implementation with metric units, extensive conversions were applied prior to the integration. In addition, the landscape plans and city furniture were also converted and integrated into the final model. The designed model included buildings, textures from design libraries for all model elements, vegetation, city furniture, transportation structure, and other units such as waste bins, Wi-Fi equipment, etc. The third stage involved implementation of different visualization platforms for different LoDs and performance optimization.

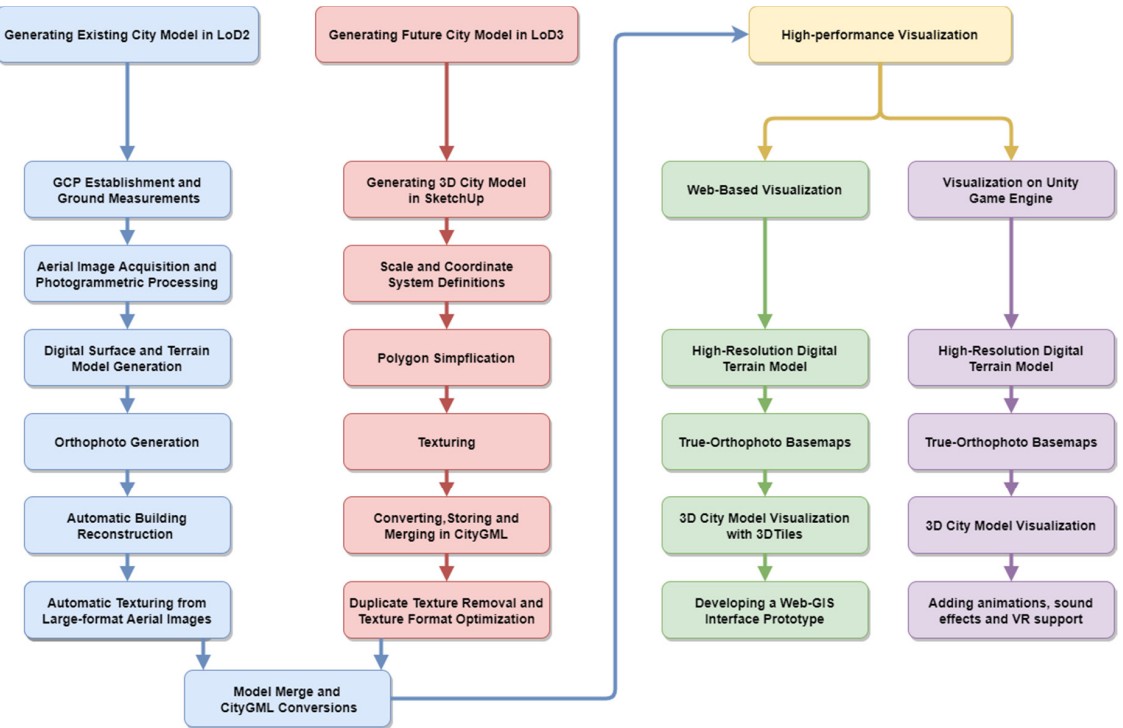

**Figure 2.** The overall workflow of the data production, model implementation, and visualization.

For the first stage, a total of 736 aerial photos taken in 2015 with 80% forward overlap and 60% lateral overlap using an UltraCam Falcon large-format digital camera from Vexcel Imaging, Graz, Austria [96] were provided by the MoEU. The photos were taken from an average altitude of 2500 m from mean sea level (MSL) with an average ground sampling distance (GSD) of 10 cm ± 2 cm, which covered an area of 83 km$^2$. The photogrammetric data production consisted of flight planning, ground control point (GCP) establishment, image acquisition, and aerial triangulation. The exterior orientation parameters (EOPs) were measured using global navigation satellite system (GNSS) receivers and an INS (inertial navigation system) in the TUREF/TM36 reference system defined with European Petroleum Survey Group (EPSG) Code 5256 during the flight mission. The aerial triangulation was performed with Trimble Inpho software, and images were provided with the adjusted EOPs and the camera calibration data by the MoEU. According to the national mapping regulations of Turkey, the photogrammetric triangulation should have an accuracy of better than 0.75 times the GSD in planimetry and 1 GSD in height. The digital surface model (DSM) of the study area with a 30-cm grid interval and digital orthophotos (10 cm GSD) was generated using Agisoft Metashape Professional, Agisoft LLC, St. Petersburg, Russia [97]. Based on the literature, the DSM is expected to have an average point positioning accuracy of ca. 1 GSD (i.e., ~10 cm) in open terrain.

The DTM with a 1-m grid interval was produced by filtering the generated DSM using lastools software (rapidlasso GmbH, Gilching, Germany) [98]. Ground filtering is a widely used method for DTM production from point clouds. The "lasground" module of lastools classifies the point cloud data as ground and non-ground points and converts ground points into a DTM. The quality of the

generated DTM depends on the DSM quality and the step size parameter used in filtering. Several step size parameters were tested for finding the optimal value.

In addition, true orthophotos generated by the MoEU using Trimble Inpho software (Trimble Inc., Sunnyvale, CA, USA) [99] were also provided for the project. The building footprints defined in the TUREF/TM36 projection system with semantic data were obtained from the GIS Department of Sahinbey Municipality, Gaziantep, in ESRI Shapefile format and exported to the CityGML format. The 3D city model in LoD2 was generated using the method proposed by Buyukdemircioglu et al. [37] with the DSM, the DTM, and the building footprints. In another study area, 73.8% of automatic building reconstruction accuracy was obtained for LoD2 model production with this method [37]. Although the building footprints and the resolution of the DSM were adequate for model reconstruction, a DTM with at 1-m grid spacing was sufficient for clamping buildings to the terrain and precisely calculating building heights from the DTM. The generated models were automatically textured using CityGRID software (UVM Systems, Klosterneuburg, Austria) [18] with a similar approach to that in Reference [37]. Texturing improves the attractiveness of a 3D city model dramatically. Georeferenced aerial oblique [100] and nadir [37] images can be used for texturing building roofs and façades, and mobile images can be used for texturing street-side façades. The texture images were converted from PNG (Portable Network Graphics) to JPEG (Joint Photographic Experts Group) format for reducing the texture data size and for increasing the visualization performance and stored in CityGML.

In the second stage, the future smart city design process was carried out using various CAD software. Highly detailed LoD3 building models, city plansm and city furniture were designed using Trimble SketchUp software (Trimble Inc., Sunnyvale, CA, USA) [101]. Several problems encountered during the conversion are given in more detail in Section 4.2, together with the solutions and the results. The generated models were imported into Autodesk 3ds Max software (Autodesk, San Rafael, CA, USA) [102] as an intermediate platform for polygon optimization, scaling, georeferencing, texturing, and CityGML conversion operations. The number of polygons is an important factor that directly affects the visualization performance of the produced models in a real-time rendering environment like a game engine or web interface (the environment where the graphics card should render the entire model vividly and fluently). The model should be represented with a small number of polygons since each surface brings additional load on hardware, particularly on the GPU (graphics processing unit). In addition, the texture quality can be more important than the number of surfaces for representing the details of the model. The texturing process was also manually carried out in 3ds Max using the standard (default) channel for lossless conversion to CityGML. The final models were firstly converted to CityGRID XML (Extensible Markup Language) and then into CityGML format. All textures were compressed in JPEG format by converting from PNG with FME software (Safe Software, Surrey, Canada) [103]. Duplicates in textures were removed and merged into atlases and stored in CityGML. For the CityGML conversion, the CityGRID Modeler plugin, which is a tool for editing, exporting, georeferencing, and updating 3D city models created in 3ds Max, was employed. A Binary Large Object (BLOB) converter tool for converting any geometry object in 3ds Max to CityGRID XML data structure was used. The BLOB converter scans the file for objects of editable meshes and transforms them into CityGRID objects.

In the last stage, all generated models were merged and visualized on the web using CesiumJS virtual globe (www.bizimsehir.org/model) and the Unity game engine (www.bizimsehir.org/unity). Virtual globes are used not only by professional users, but also by public users due to their intuitive interface and web-based system [104]. Among the available open-source web globes, Cesium runs on a GPU using WebGL and supports various geospatial data types, such as terrain, imagery layers, and 3D geometries with high LoD and textures. The implemented 3D city model with buildings, ground plans, and city furniture was also exported to the Unity game engine to provide a different visualization experience to the users and stakeholders in an urban simulation environment with VR. Due to the various optimization approaches used for the different visualization approaches, the methodological details on this part are elaborated on further in Section 5 together with the results.

## 4. Model Implementation Results

### 4.1. Semi-Automatic Model Generation of Existing 3D City Model in LoD2

The 3D city models of the three neighbor districts consisting of 1202 buildings were reconstructed semi-automatically in LoD2 using photogrammetric techniques and large-format aerial images. Semi-automatic 3D city model reconstruction was performed with BuildingReconstruction software (virtualcitySYSTEMS GmbH, Berlin, Germany) [105]. The software uses a model-driven approach for 3D city model reconstruction, and it allows users to perform manual editing on the generated models. After visual assessments, no manual editing was required in this study due to high-quality output. The BuildingReconstruction software produces CityGML LoD2 3D city models using DSM, DTM in raster format, and building footprint data as input. Attribute information, such as building name, building condition, number of apartments, usage type, etc., obtained from the building footprint file, was stored as semantic data in CityGML together with the geometries to perform queries.

The compatibility of the building footprint data provided by the municipality with the true orthophotos provided by MoEU was visually compared using a simple overlay. The missing building footprints were manually digitized from orthophotos, and the footprints of the newly constructed buildings were deleted to ensure visual consistency of the model. A general view of the building footprints and the borders of the modeled neighborhoods (three sub-districts) is shown in Figure 3.

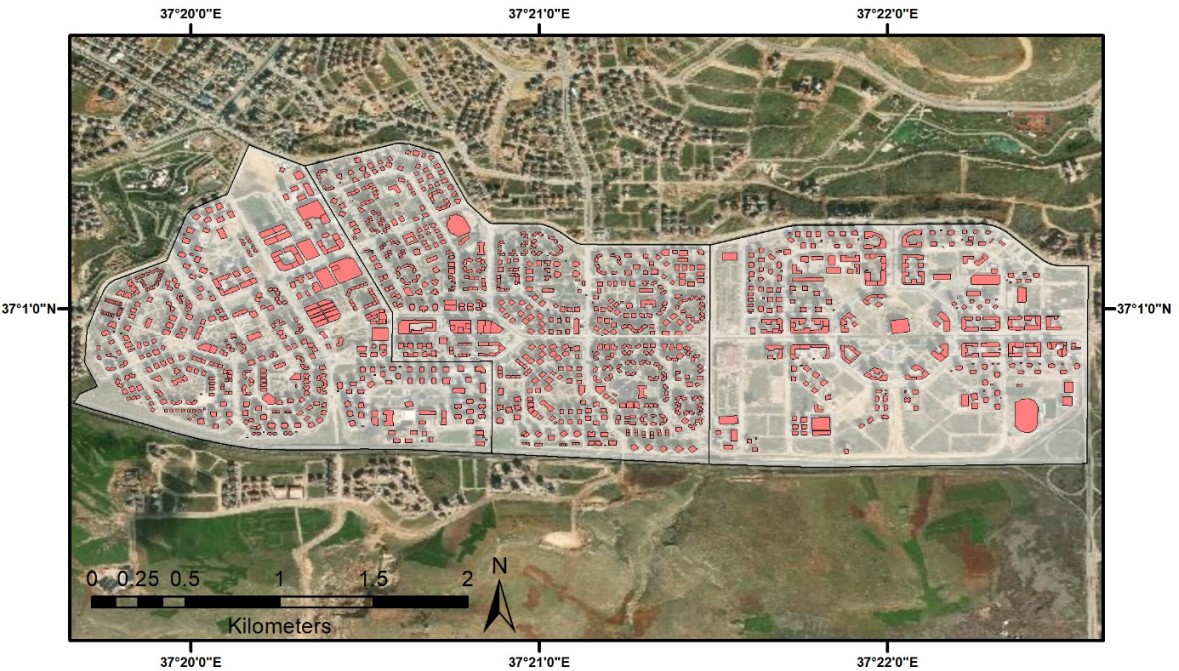

**Figure 3.** A general view of the boundaries of the modeled city districts and the building footprints located in the north of the project site.

An overview of the generated DSM and the DTM obtained with ground filtering can be seen in Figure 4. The results demonstrate a smooth DTM with few errors in the locations of large buildings, as marked by the red polygon.

Unity Game Engine visualization of the textured LoD2 model on DTM with the true-orthophoto basemap is shown in Figure 5. As can be seen from the figure, the reconstructed LoD2 model aligned well with the DTM and the orthophoto, which demonstrates the consistency and high geometric accuracy of the building geometries and textures with the related data.

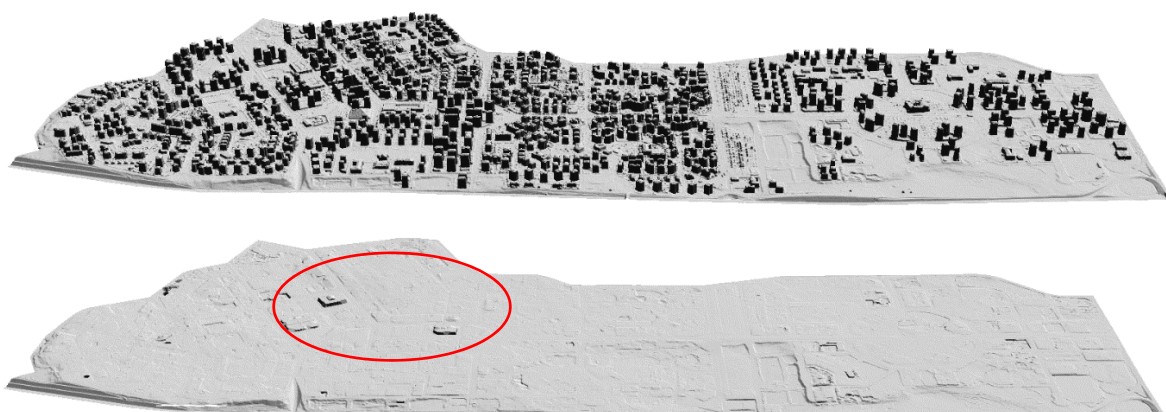

**Figure 4.** An overview of the generated digital surface model (DSM) and the digital terrain model (DTM), showing one area (red) with DTM filtering errors.

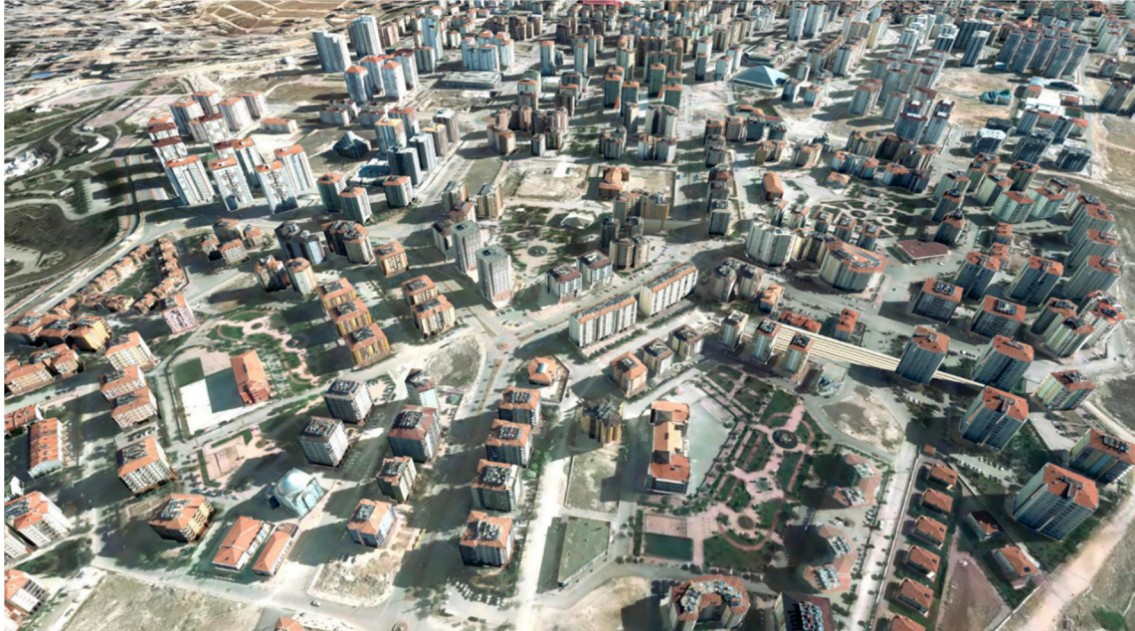

**Figure 5.** Unity Game Engine visualization of the textured LoD2 model together with the DTM and the true-orthophoto basemap.

## 4.2. Design and Model Generation of the Future Smart City Model in LoD3

SketchUp software is suitable for rapidly generating highly detailed 3D city models, but some post-processing was required since the models were not georeferenced and did not fit to the urban plans. In addition, due to the very high number of polygons involved in the models, the number of polygons for each building needed to be optimized by manually eliminating the unnecessary ones. In the elimination process, it was considered not to cause any deformations to the visual details of the models. Through manual editing, the topological correctness of the model geometries was also ensured, and each object was assigned with a unique identification number. An example of the building models designed with the SketchUp tool is shown in Figure 6. Figure 7 shows the initial and optimized wireframe models of a building.

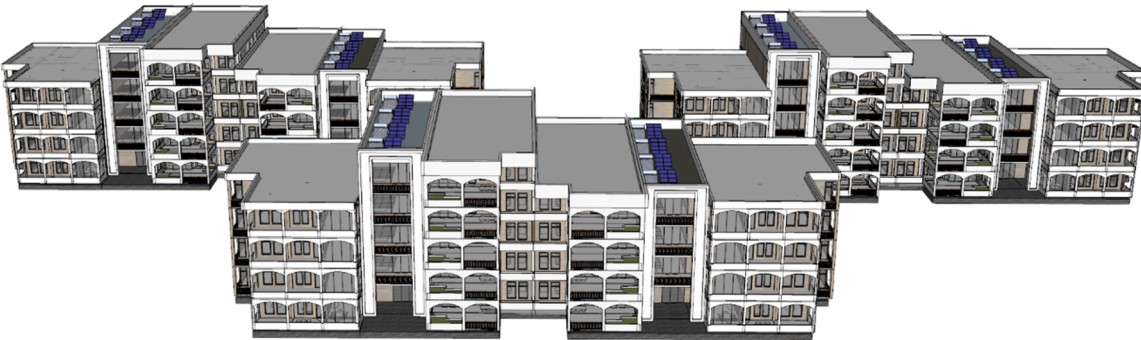

**Figure 6.** Example of building models designed in the SketchUp software.

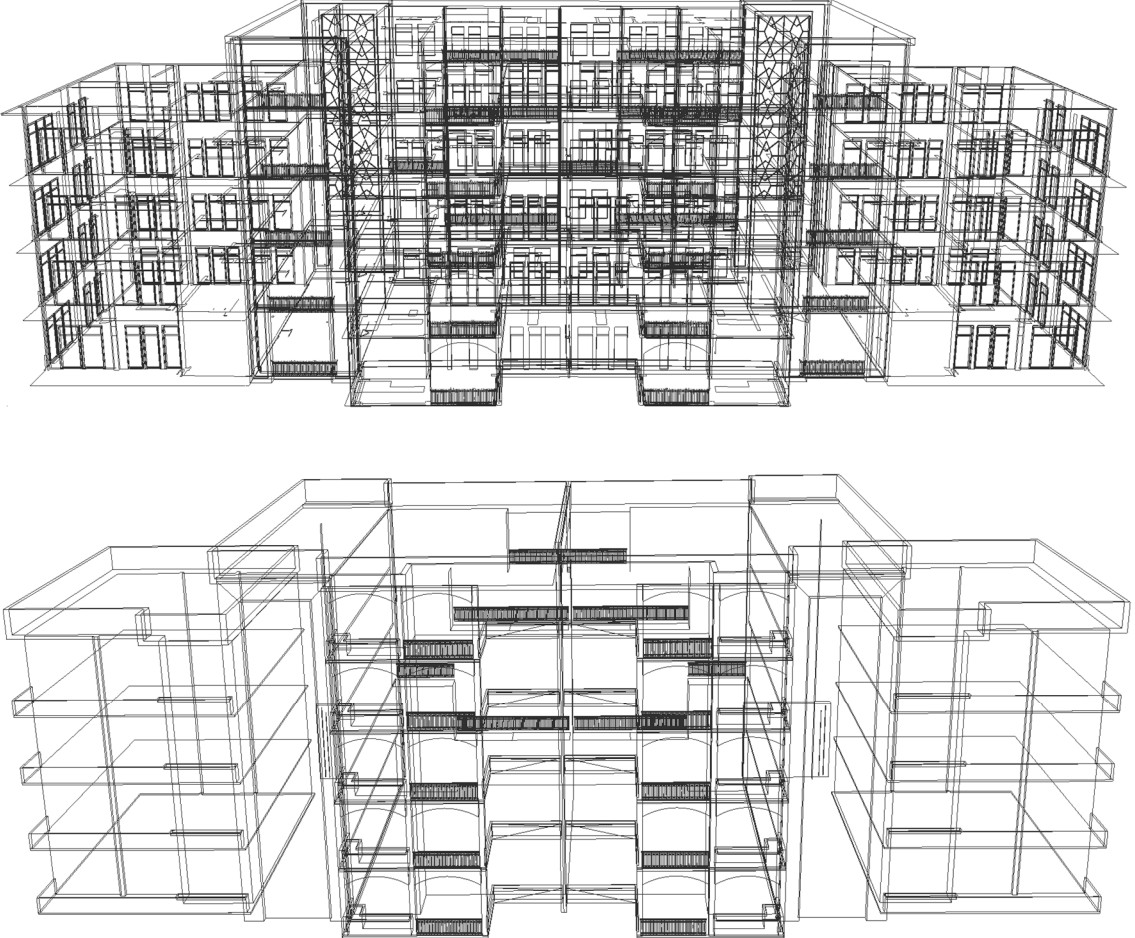

**Figure 7.** A non-optimized building model with 1,278,000 polygons (**top**) and an optimized building model with 24,000 polygons (**bottom**).

Scaling and metric coordinates were other issues faced in the model production. Although the georeferenced urban plan was developed first, the architectural designs were not metric and had some differences in comparison to the planned building footprints. The main reasons were the choice of the design software by the architects and modifications made in the detailed architectural designs. Most buildings were designed as individual units; they were also enlarged, shrunk, and rotated by the architects to work comfortably, and they were without georeferencing. Since the 3ds Max software does not support any projected coordinate system, it was not possible to directly use the models in the city model context. Another reason was that the 3ds Max software supports up to eight-digit coordinates, including decimal characters. Therefore, when modeling with 3ds Max, an arbitrary local coordinate

system was used with reduced coordinates by "shifting the plan to zero", to limit all values to eight digits at maximum. In order to georeference the models precisely, the 2D urban plan developed by the urban planners was employed with the same reduction (offset) values, and the 3D building models were placed manually on the urban plan. A part of the urban plan [16,106] can be seen in Figure 8. A close view of a building model before and after texturing is shown in Figure 9.

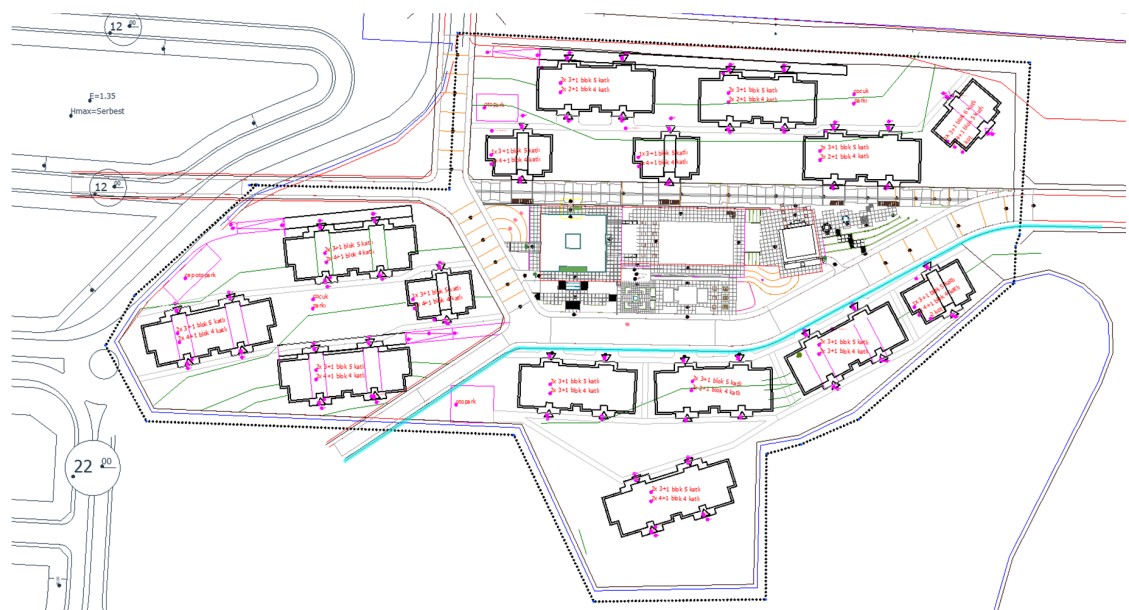

**Figure 8.** A part of the project site plan with georeferencing information in two dimensions (2D) [16,106].

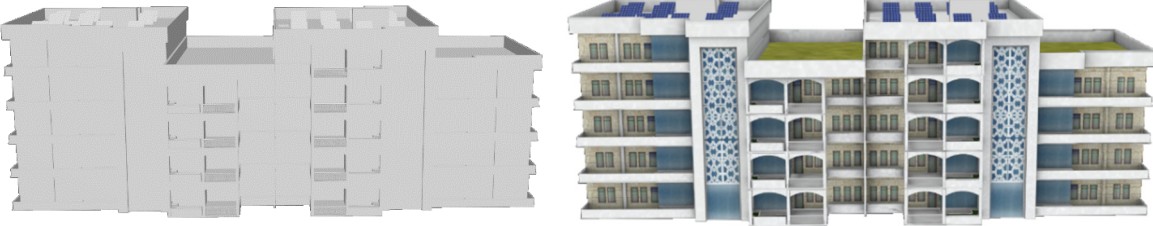

**Figure 9.** A close-up view of building model before texturing (**left**) and after texturing (**right**).

The surface normal (or normal vector) is another important factor in model generation that identifies the surface direction (front or back face). The direction of the vector depends on the order of the vertices and coordinate system definition (right- or left-handed). The front face of the model surface is the direction of the thumb. In the real-time visualization of models such as in a game engine, the surfaces can be shown as two-sided by doubling the surfaces, which brings more load to the graphics card. If the surface normal faces the right, a stereo view is created because the graphic card shows two surfaces in the same position. The face normals can be checked in 3ds Max using the FaceNormal function or rendering. In order to avoid problems that may occur while visualizing the model in a game engine, architectural models need be produced following the requirements of real-time visualization in terms of data size (e.g., small number of polygons, small texture files, etc.).

Buildings and the urban design plan with city furniture were converted and saved in CityGRID XML format in separate files; then, each part was exported to CityGML with the CityGRID Modeler plugin. Before exporting models to CityGML, some parameters like coordinate offsets and texture quality must be set by the user. The CityGRID plugin can also generate semantic data automatically for each building with options such as 2D Area, LoD1 Height, LoD2 Eave Height, LoD2 Ridge Height, LoD2 Roof Area, and Roof Type. The BLOB conversion parameters and CityGML export parameters are given in Figure 10. A part of the future city model in 3ds Max is shown in Figure 11.

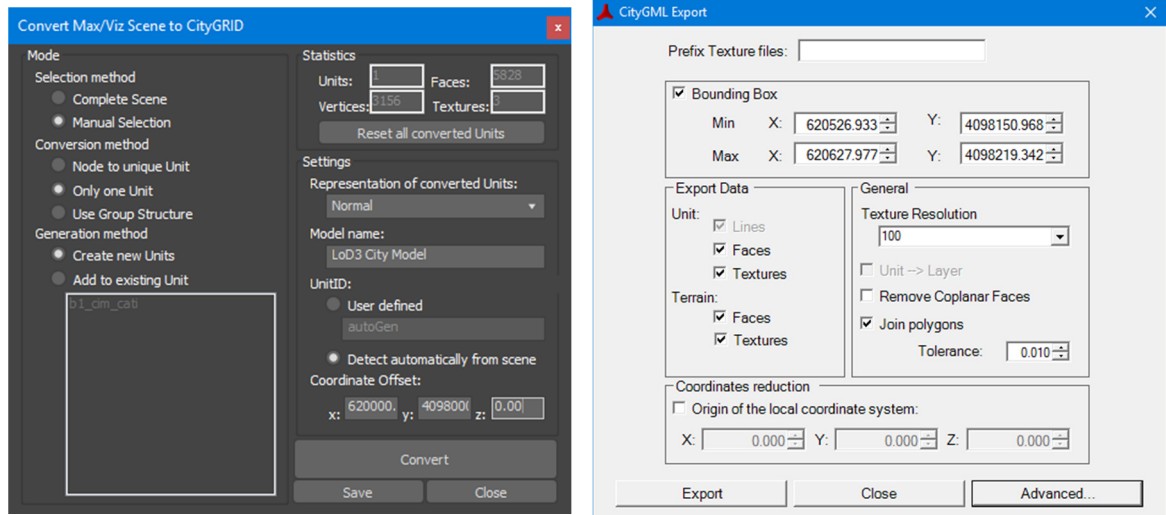

**Figure 10.** The Binary Large Object (BLOB) conversion parameters (**left**) and CityGML export parameters (**right**).

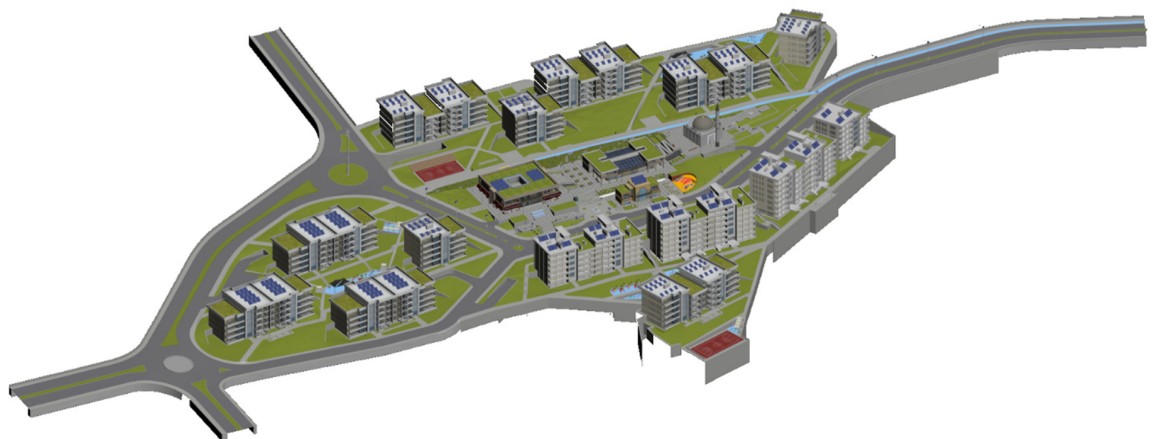

**Figure 11.** A part of the future city concept model in 3ds Max tool.

## 5. High-Performance Visualization Results

### 5.1. Web-Based Visualization Using CesiumJS

According to Chaturvedi et al. [85], the main features of Cesium are support for visualizing 2D and 3D data (vector and raster) and web map services (WMS) on a web-interface, navigation with different camera angles and zoom levels, customization of object façade symbology, support for widely used reference systems, and dynamic visualization. Based on the experiences obtained during this study, additional useful features of Cesium can be listed such as interface customization, utilization of the 3D Tiles format for streaming, styling, and interaction possibilities, visualization of high-resolution DTMs globally, creating info boxes for object selection and attribute display, enabling dynamic visual effects for atmospheric changes, shadows, fog, sun light, etc., and performance optimization for reduced GPU, CPU (central processing unit), and power utilization.

The CityGML data were converted into 3D Tiles format for high-performance visualization on the web. The 3D Tiles format is a general specification for visualizing large heterogeneous 3D geospatial datasets such as buildings, textured meshes, or point clouds on CesiumJS virtual globe. It also supports rendering large volumes of datasets as tiles. Through tiling, the amount of hardware resources used by web browser is reduced and the streaming performance is increased. The tiles were selected based on geometric error for detail-level and an adjustable pixel defect, so that multiple zoom levels in

the same view could be obtained with high performance. 3D Tiles stores the tile geometries in glTF (Graphics Library Transmission Format) [107] format, which is the de facto format for 3D visualization applications, as well as for CesiumJS, and which can also store semantic data. The advantages and disadvantages of 3D Tiles experienced in this study are given in Table 1.

**Table 1.** Advantages and disadvantages of 3D Tiles as experienced in this study.

| Advantages | Disadvantages |
| --- | --- |
| • High performance while visualizing large and complex datasets.<br>• Storage of geometries and textures in the same file.<br><br>• Lossless compression (Gzip) support for reducing file size.<br>• Picking, styling, and querying on dataset. | • Lack of support for visualizing the data directly from a geodatabase.<br><br>• No online update possibility on the attribute data.<br><br>• Limited availability of open-source tools for CityGML to 3D Tiles conversion. |

Cesium ION [108] is a cloud-based hosting platform, which was also employed here, for optimizing, tiling, and serving the geodata, i.e., the 3D city model, high-resolution DTM, and true orthophotos. The visualization options investigated here are described below.

### 5.1.1. High-Resolution Terrain Visualization

A 3D city model without a high-resolution DTM would also be incomplete and may provide misleading information. If a sparse DTM is used with the city models, parts of the building models may sink in the terrain or could hang in the air. Therefore, a high-resolution DTM (1 m or better) coherent with the building models is essential to avoid visual deformations. CesiumJS is capable of visualizing and streaming high-resolution DTMs in different formats. There is, however, support for one DTM at a time, and a single file must be formed even for large regions. Terrains can be visualized as regular grids or triangular irregular networks (TINs) in CesiumJS with the support of Heightmap 1.0 [109] or Quantized-mesh 1.0 [110] formats, respectively. The Heightmap 1.0 employs regular grids in multi-resolutions and adjacent tiles, which also have a small overlap in order to ensure model continuity. The Quantized-mesh 1.0 format pre-renders a TIN for each tile and can be better optimized for different terrain types, such as by providing more details for rugged terrain surfaces. A visual comparison of both formats is provided in Figure 12.

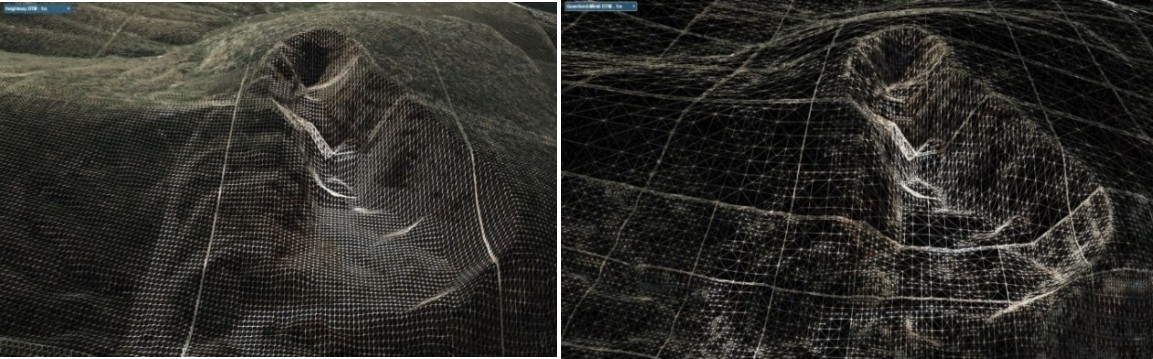

**Figure 12.** Regular and irregular representations of the terrain with Heightmap 1.0 (**left**) and Quantized-mesh 1.0 (**right**) on Cesium.

In this study, Cesium ION platform is employed for generating and serving the Quantized-mesh terrain tileset as GeoTIFF, which is among the supported raster file formats [111]. A number of raster pre-processing steps were performed prior to the upload for seamless conversion, such as conversion to a single band of floating point or integer elevations, which must be defined with respect to either

MSL (i.e., EGM96) or WGS84 (World Geodetic System 1984) Ellipsoid. The Cesium ION platform can combine uploaded terrain datasets with the Cesium World Terrain (CWT) to create a new global terrain layer. CWT is a global coverage terrain model with several resolutions. Cesium ION Terrain Tiler replaces the area of the uploaded terrain model from the CWT and fuses it into a single Quantized-mesh terrain tileset optimized for efficient streaming into CesiumJS and other 3D engines. An example of CWT merged and unmerged with the high-resolution DTM produced in this study is shown in Figure 13 together with the textured LoD2 model.

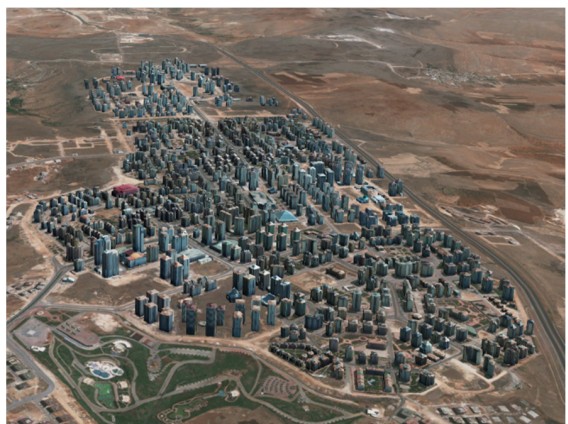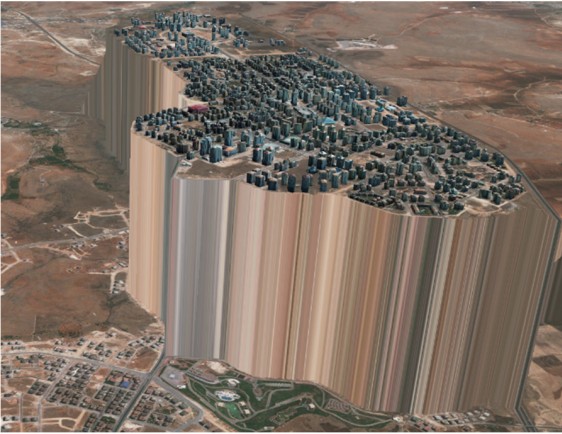

**Figure 13.** An example of Cesium World Terrain merged (**left**) and unmerged (**right**) with the high-resolution DTM and with the textured LoD2 model in the study area.

A pre-processing step on the DTM of the designed future city was applied in order to conform the plans with the existing city parts, since using the original DTM for the project area would yield incompatibility and cause visual deformations. The project area was removed from the existing DTM and filled with a new DTM that complies with the designs using the FME software, implementing the Clipper, RasterCellValueReplacer, and RasterMosaicker transformers. A part of the designed city model with the high-resolution DTM before and after pre-processing can be seen in Figure 14.

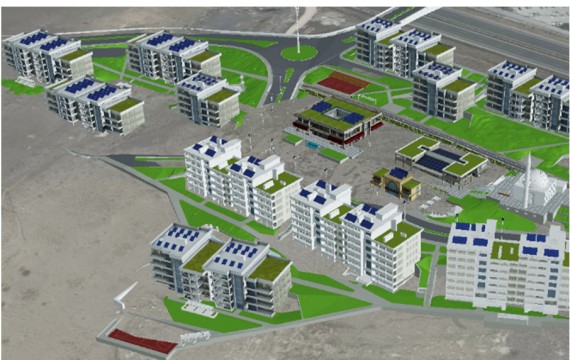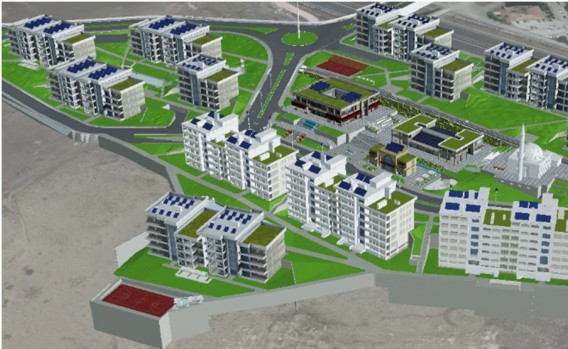

**Figure 14.** The planned city model with the original (**left**) and pre-processed (**right**) DTM.

5.1.2. Basemap Generation Using True Orthophotos

High-resolution true orthophotos should be used as basemaps for fine visualization of 3D city models to provide a realistic impression and for accurate positioning of the models on the virtual globe. True orthophotos provide better radiometric and geometric quality, since standard large-scale city orthophotos exhibit problems due to the displacements and occlusions [112]. High-resolution orthophotos were produced as part of the photogrammetric processing workflow, and the true orthophotos with the same resolution were compared visually. The true orthophotos were found superior due to smaller visual deformations especially at roof edges (Figure 15).

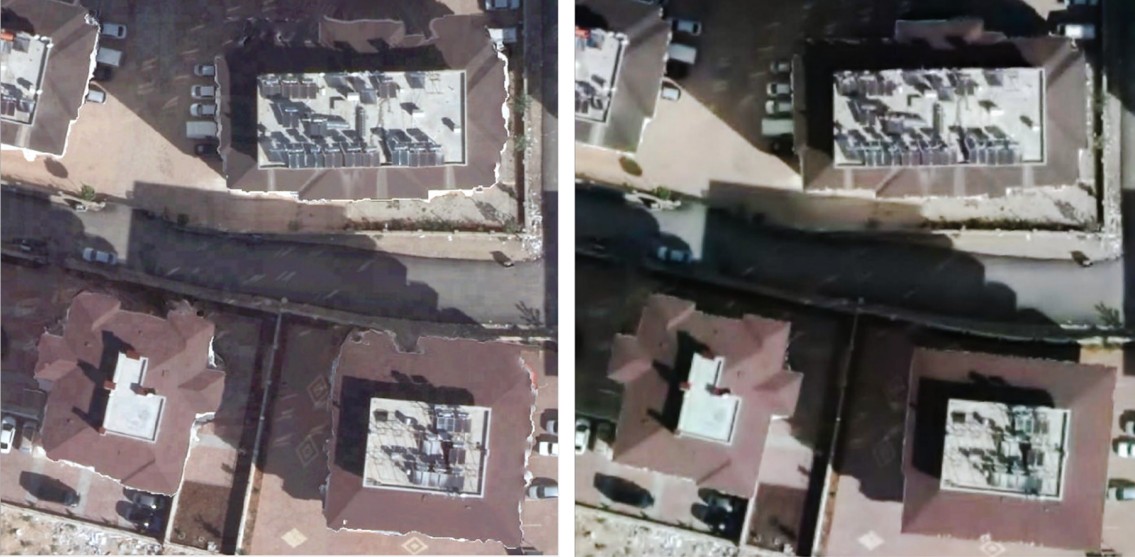

**Figure 15.** Radiometric comparison of ordinary orthophoto product (**left**) and the true orthophoto (**right**). The differences are especially noticeable at the roof edges.

The true orthophoto files were tiled into nine GeoTIFF files, JPEG-compressed by 25%, and reprojected to the Web Mercator projection (EPSG:3857), which is the default of Cesium ION; then, they were uploaded to Cesium ION and finally converted into TMS and Web Map Tile Service (WMTS) imagery layers. When uploading multiple overlapping tiled imagery, all raster files must have the same resolution in order to avoid resolution conflicts within the dataset. As a last step, true orthophoto basemaps were compared visually with various basemap imagery layers from other data providers using DTM and building models, as shown in Figure 16.

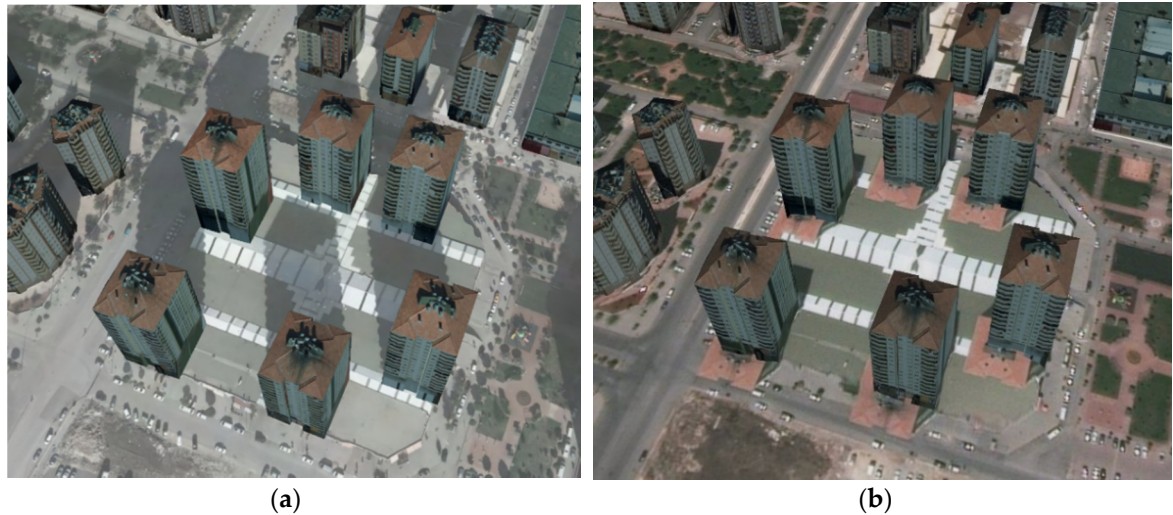

(**a**)　　　　　　　　　　　　　　　　　　　(**b**)

**Figure 16.** *Cont.*

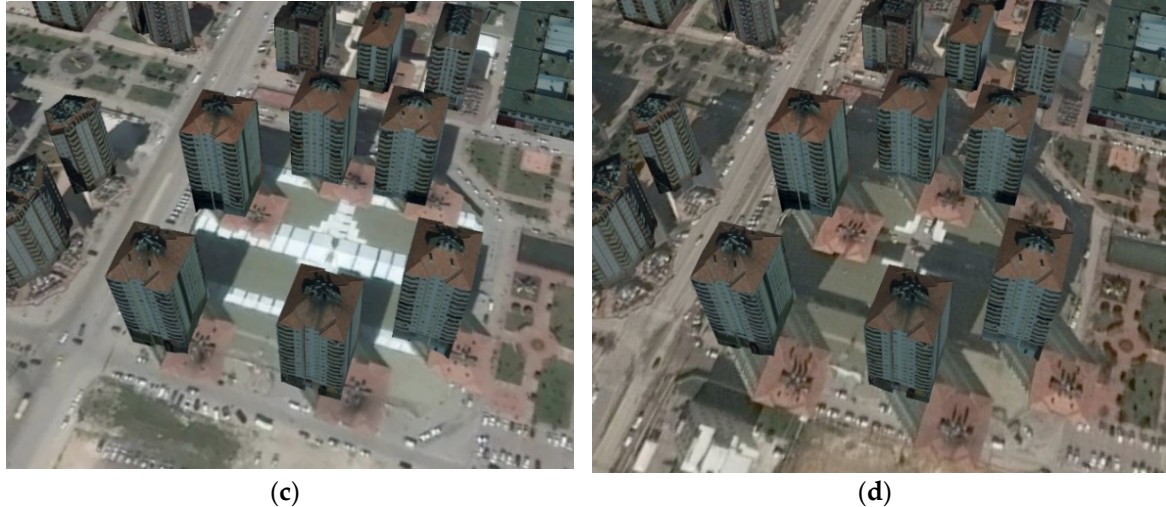

|（**c**）|（**d**）|

**Figure 16.** Three-dimensional buildings visualized on (**a**) true orthophoto basemaps, (**b**) Bing Maps, (**c**) Mapbox Satellite, and (**d**) ESRI World Imagery.

### 5.1.3. 3D City Model Visualization with 3D Tiles

Prior to the conversion into the 3D Tiles format, a number of pre-processing and optimization steps were applied to the CityGML data for increasing visualization performance and lowering the bandwidth usage. In the LoD3 future city model, the building and city furniture models, as well as the urban plan, were stored in separate CityGML files along with their textures. Textured CityGML files may contain thousands of separate image files, which can easily reach hundreds of megabytes just for a couple of buildings. There were redundant texture data in the CityGML files that caused increased file sizes. Therefore, models with common textures were merged into a single CityGML file and the duplicates were removed. Two separate CityGML files with textures were created for buildings and landscape designs with furniture. These files were merged using the 3DCityDB software, which is an open-source software package for managing CityGML-based 3D city models developed at Munich Technical University, Munich, Germany [113,114].

Texture atlas is an image mosaicking method used to reduce the data size [115]. Atlased textures involving multiple facades were used for texturing buildings (Figure 17). When atlases are used instead of requesting a texture file for each façade, the hardware usage, rendering speed, and streaming performance can be optimized dramatically. A comparison of the number of textures and file size of the model before and after the optimization is given in Table 2.

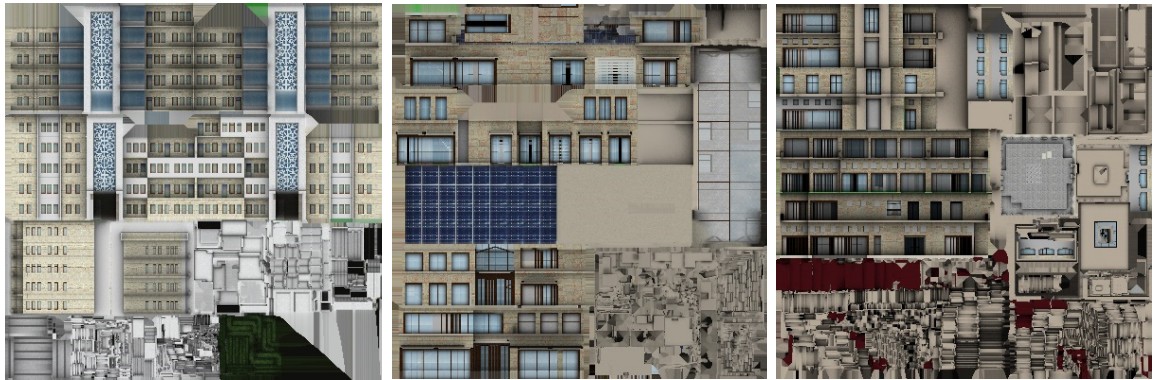

**Figure 17.** Examples of atlased textures of the building models.

**Table 2.** Comparison of the number of the texture images and the total file sizes before and after optimization.

| LoD2 Models PNG Textures | LoD2 Models JPEG Textures | Landscape Plan and Furniture (Before) | Landscape Plan and Furniture (After) | LoD3 Buildings (Before) | LoD3 Buildings (After) |
|---|---|---|---|---|---|
| 15.454 files | 15.454 files | 608 files | 33 files | 3.847 files | 154 files |
| 709 MB | 100 MB | 1.233 MB | 83.1 MB | 568 MB | 94.4 MB |

Although there are open-source tools for converting CityGML to 3D Tiles, only Cesium ION provides lossless conversion, and it was used here for the conversion. 3D Tiles is a hierarchical tile format, and the streaming performance depends on the geometry and rendering optimizations. The smaller data size allows for faster streaming and rendering on the web interface. The lossless gzipping support of 3D Tiles also increases rendering, streaming, and runtime performance and improves the efficiency and the speed of transmitting the 3D content over the web. Draco compression is another solution, which compresses vertices, normals, colors, texture coordinates, and any other generic attributes [116]. The models had smaller data size after the compression and were streamed faster using 3D Tiles due to its ability to stream glTF models when new tiles come into view or a new LoD is required.

In this study, the LoD2 model was enriched with attributes coming from GIS database, and the LoD3 future city model had few semantic data, such as height, terrain height, latitude, longitude, and number of rendered primitives, which were automatically generated from geometries by Cesium ION while converting from CityGML to 3D Tiles without any loss. Improper attribute data types (string, integer, double) prevent the functioning of querying or styling on the generated models. 3D Tiles Batch and feature tables were employed for semantic data conversion. A web interface prototype was developed on which the buildings and other types of objects can be selected, styled, and queried, such as grouping the objects with respect to their attributes (e.g., coloring buildings with longitudes as shown Figure 18). An example of building attributes displayed on the web interface is shown in Figure 19. Additional functionality such as mensuration of building heights, terrain elevation, and angles on the displayed model was also developed in the prototype (Figure 20).

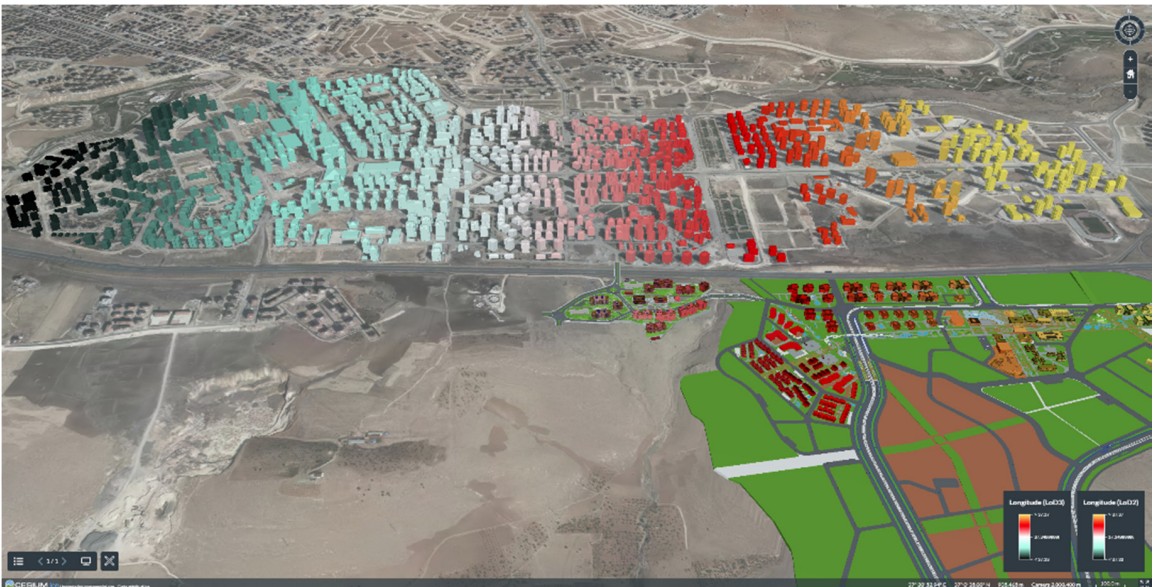

**Figure 18.** LoD2 building models model colored with their longitudes using 3D Tiles styling.

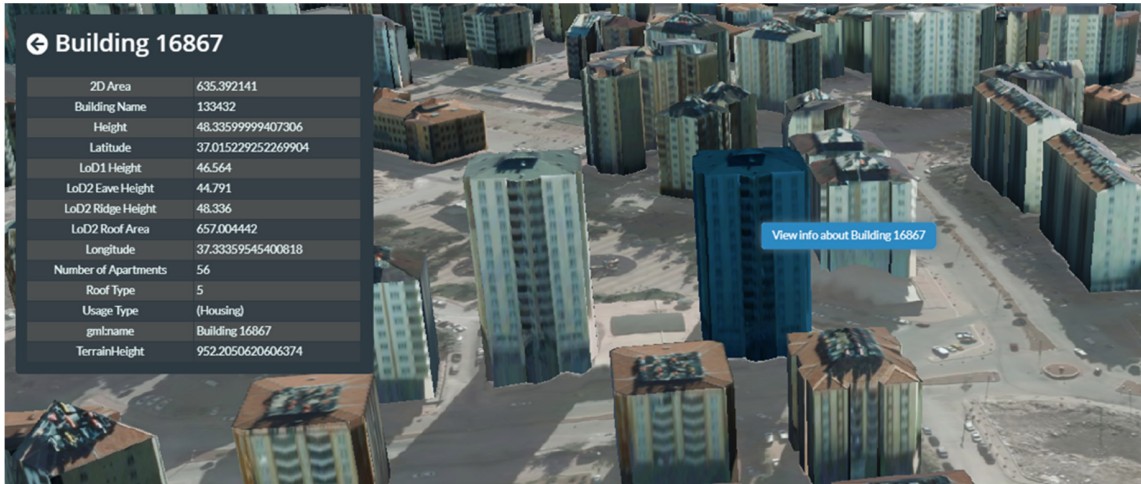

**Figure 19.** Displaying selected LoD2 buildings attribute table on the customized web interface.

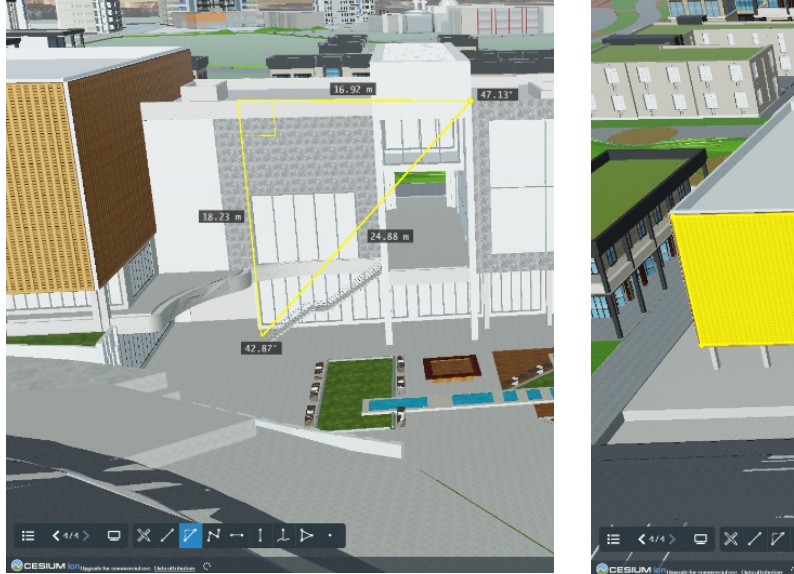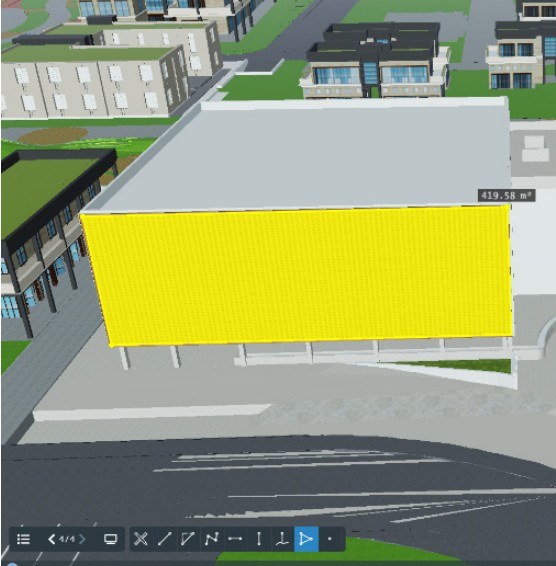

**Figure 20.** Component distance (**left**) and area measurement tool (**right**) on the developed web interface.

## 5.2. Exploring Smart Cities with Virtual Reality in Unity Game Engine

All objects generated in 3ds Max could be converted into Unity with their position, rotation, and scale information using Autodesk FBX [117,118] without any loss, since the model implementation stage (georeferencing, scaling, etc.) was performed considering the Unity requirements as well. The vertex colors, normals, and textures were exported as multiple materials per mesh. Sound and moving effects, such as people talking, trees swaying in the wind, etc., were also added in order to create a lively urban simulation. Such approaches can be useful for the improved interpretation of an environment, such as understanding the level of city noise at a certain location by hearing in the simulation. A view of generated LoD3 model in the Unity game engine can be seen in Figure 21.

The VR technology requires both special hardware and software for exploring the model such as a VR-ready GPU and VR headset. In this study, the generated urban simulation was explored using the HTC Vive [119] virtual reality headset, SteamVR [120] software, and GeForce GTX 1080 GPU. The scene was produced in two different versions: one for exploring model with the combination of keyboard and mouse and one was for exploring with the VR headset (Figure 22).

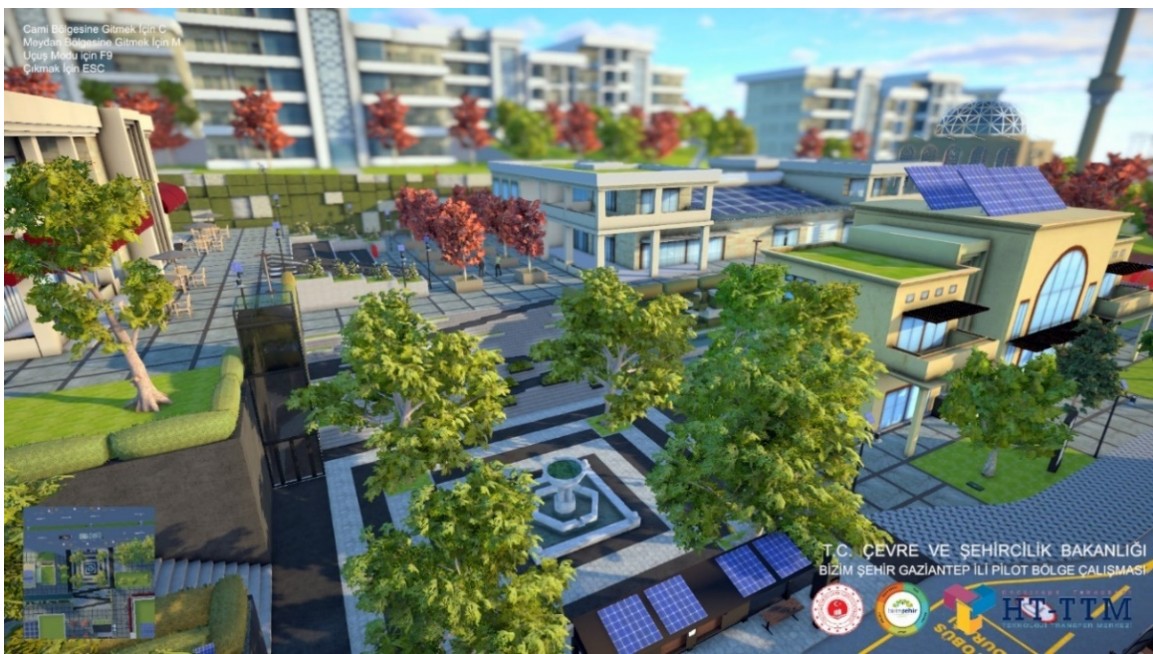

**Figure 21.** Unity simulation of the LoD3 future city model generated in the study.

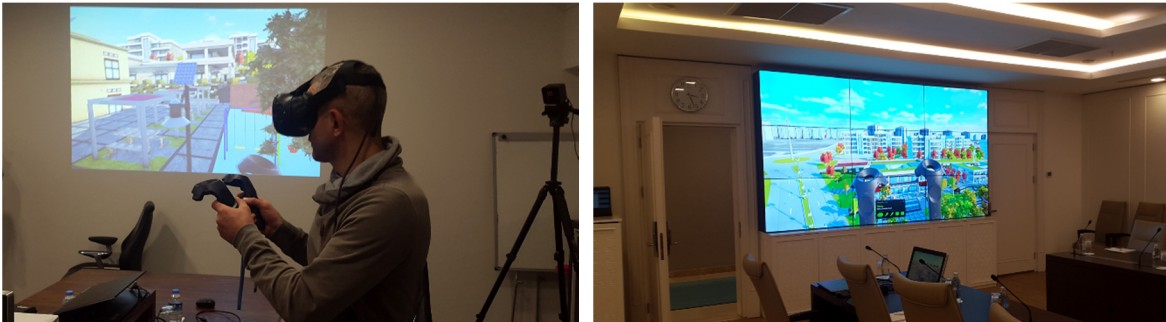

**Figure 22.** Virtual reality exploration of the LoD3 future city model using the Unity Game Engine environment.

## 6. Discussion

The present study showed that the generation of highly detailed 3D city models is a challenging task in many aspects and requires expertise with various software and extensive manual effort. Aerial photogrammetric techniques provide the most suitable and cost-effective data for generating the LoD2 3D city models semi-automatically. Basemaps, high-resolution DTMs, and building textures can also be generated from the large-format aerial images. Parts of these tasks, such as high-resolution DTM and building geometry extraction, can also be accomplished using multi-view very-high-resolution satellite imagery, but with less geometric detail and lower texture quality.

Integration of the models coming from different sources has certain challenges, such as differences in resolution and accuracy, topological consistency, etc. This study could overcome these issues since both models were located in different parts of the terrain and merged through the high-resolution DTM. The LoD2 model had a geometric accuracy of ca. 10 cm coming from the aerial images and photogrammetric triangulation. The designed buildings needed to be edited extensively to reduce the data size and ensure topological consistency within the models. They were georeferenced by scaling and manually fitting to the urban plan. Thus, visual and topological consistency could be ensured in the area by using high-resolution and accurate datasets. The importance of DTM quality must be particularly emphasized here, since it is the connecting medium for the models.

High-performance visualization of the 3D models often requires post-processing for geometry optimization; thus, the hardware requirements and the bandwidth usage can be reduced for rapid visualization by decreasing the data size. Using texture atlases for this purpose is a solution that is recommended for reducing the data size. Integrating the existing digital cities with the planned models and using joint visualization platforms, such as Cesium JS, Unity, etc., can facilitate participatory planning and decision-making by providing realistic 3D representations of the designs in the environment that they belong to, which can assist discussions by ensuring improved visual experience and understanding of the designs, as well as help to present the developed concepts and designs to the public in order to collect their opinion for potential improvements. However, if the platform has weaknesses such as poor streaming and model exploration performance, low geometric accuracy and visual correctness, inconsistent data, and a lack of querying and analysis options, the usage would be limited.

Open-source CesiumJS was found to be a suitable tool for visualizing the 3D city model, terrain, and basemaps on the virtual web globe, while also relating them to the 3D GIS environment. It provides the capability of switching between different datasets in different LoDs to the users. In addition to the building models and semantic information, the 3D Tiles format provides a good solution for various applications. The city models in CityGML format can be directly converted into 3D Tiles without losing any geometric and semantic information. Instead of loading the entire city model at once, tile-based model loading provides users higher performance on the web interface and reduces hardware load on the computer.

Currently, the main issue in web-based visualization of 3D city models using open-source software is the model updating. Since the visualized model is static and not directly visualized from a database or a similar dynamic source, the whole model must be generated again every time there is an update on the model. Serving a city model with topography directly from a spatial database management system (DBMS) to the web interface would eliminate this problem. Efficient geospatial database solutions should be developed for high-performance visualization of city and terrain models, which is of great importance for updating such models.

It was observed throughout the study that using game engines and VR technology has certain advantages, such as joint provision of an improved visual experience on the designed and the real environments. Using visual effects in the created scene also provides a lively feeling. Game engines mainly use a GPU, enable visualization of more detailed scenes, and provide higher FPS (frames per second) compared to web-based visualization. Using VR with game engines gives users a more realistic exploration of the model with animations, sound effects, and better interaction possibilities with the model. Exploring the model at the street level is also possible. On the other hand, the scenes created in Unity are standalone tools and cannot provide the advantages of a 3D webGIS platform, such as querying, online data sharing, using different basemaps and other web based data services, spatial analysis, semantic data integration, etc.

## 7. Conclusions and Future Work

The outcomes of this study can be seen from the perspective of the development of a 3D GIS environment with highly detailed 3D city models, true orthophotos, and high-resolution terrain models as base data. Such systems can support the demands of smart city projects by ensuring the inclusion of professionals from various disciplines and citizens in solving problems. Development of such systems is an active research area and has several challenges, such as production, management, and updating of 3D object geometries, semantic data integration, implementation of specialized spatial queries, high-performance visualization, cross-platform accessibility, geometry validation procedures, etc. Photogrammetry and remote sensing methods can provide the data required for generating accurate and detailed geometries and for updating them regularly.

This study demonstrated the modeling stages of an integrated 3D city model of an existing city section and a designed future city in LoD2 and LoD3, respectively; it also investigated different

visualization platforms and performance optimization procedures. In the model implementation stage of existing urban areas, photogrammetric techniques can be used as the primary data source. Future cities are often designed in CAD environments and their conversion into geographical data exchange formats and applications, such as CityGML, requires extensive editing and format transformations. Lossless conversion between different systems can be a challenging task. Fusion of multi-source and multi-temporal data also requires special attention. Here, the CityGML format was preferred since semantic data, texture, and object geometries could be stored in an integrated manner and the format could be converted to 3D visualization formats for different platforms.

Web-based high-performance visualization of detailed 3D city models is still a challenging task. The CesiumJS library and Unity game engine were implemented and compared for different aspects, such as sense of realism for the users, level of details in the models, visualization performances, object and data type compatibility, inclusion of semantic data, flexibility in using external data resources such as web map and web feature services, querying, styling, etc. CesiumJS was found to be a suitable platform for the development of a 3D GIS with online data access and provision of a participatory planning environment. On the other hand, Unity with VR support has more attractive visual effects and improved sense of reality for users who intend to feel the future smart city within an actual geographical context.

Future work related to the developed platform is almost unlimited. The interior designs of the buildings (LoD4) can be integrated into the models. BIM data, which can be produced with photogrammetric techniques, can also be integrated into the models and presented on both platforms. Various building models or future city concepts can be alternatively visualized in the developed CesiumJS virtual globe supported platform. Semantic data can be diversified and integrated into the generated smart city concept for developing new queries and styling methods on the web interface. In addition, a more comprehensive 3D GIS environment with an extended number of data layers for spatial analysis, environmental modeling, and simulation is possible to especially aid urban governance. Interaction possibilities for multiple users for modifying the model geometries could also be of interest for efficient communication and collaboration, which is currently not possible within the proposed framework. Finally, provision of up-to-date and accurate data is essential to ensure the continuity and usability of the developed platform, such as for monitoring the construction stages of the designed future city or for integrating recent environmental data. Remote sensing and photogrammetry data are the primary sources for data collection and validation.

**Author Contributions:** Conceptualization, S.K. and M.B.; methodology, software, investigation, data curation, writing—original draft preparation, and visualization, M.B.; validation, formal analysis, resources, supervision, writing—review and editing, project administration, and funding acquisition, S.K. All authors have read and agreed to the published version of the manuscript.

**Funding:** This research was funded by the Ministry of Environment and Urbanization, Turkey.

**Acknowledgments:** The authors would like to thank Alpaslan Tütüneken for providing CityGRID software and the Ministry of Environment and Urbanization for providing the data and funding the project. The authors are also grateful for having the possibility to collaborate with the Bizimsehir Gaziantep project team. The authors sincerely thank the anonymous reviewers for their useful comments and suggestions.

**Conflicts of Interest:** The authors declare no conflicts of interest.

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
