# Peer review of "Reconstruction and Efficient Visualization of Heterogeneous 3D City Models"

_remotesensing, doi:10.3390/rs12132128_

Round 1

Reviewer 1 Report

Reviewed paper is devoted to 3D city modelling from various data inputs. This topic is undoubtedly interesting and above all current. I see the main problem in the fact that paper is not innovative. It is just a description of the application of a huge number of software technologies. In my opinion, the authors try to describe too much of their work in one paper. This paper tries to describe the creation of two variants of the 3D models (LoD 2 and LoD 3) and two methods of 3D visualization, without being these 3D models and visualization methods somehow compared or evaluated. I am also not entirely sure how much the paper fits in the scope of Remote Sensing journal.

There are also other serious shortcomings in the paper. See the list of major comments:

Title

  • The title of the article is too general. The source of the semantic data is hardly mentioned in the text of the article, although the term "semantic data" appears in the title.

Abstract

  • It is relatively general, which is due to the number of described processing steps and the possibility of visualization.

The general structure of the paper

  • It would be better to name the sections with standard names: Introduction; Related Work; Materials and Methods; Results; Discussion; Conclusions and Future Work.
  • The paper should be significantly shortened, and some parts of the text should be moved to other subsections (see below).

Introduction

  • The introduction is quite long.
  • After reading the introduction, it is not clear to me why they converted to CityGML format was done, when the result is (only) visualizations. CityGML is suitable when using both 3D geometric and semantic information, for example in spatial analyses and simulations, or when the created 3D model is provided in the form of open data.

Background on 3D City Representation

  • Rows 140-143: If any articles that present and compare different web 3D technologies, such as Herman & Reznik (2015, http://dx.doi.org/10.5194/isprsarchives-XL-3-W3-479-2015), this subchapter could be shortened or compacted.
  • Rows 161-162: Is temporal data important for the practical part of this paper? If not, this sentence could be deleted.
  • Rows 164-167: CityGML was primarily created for noise mapping, this use is not mentioned here.

The Study Datasets and the Main Methodological Workflow

  • Figure 1 - the text of the map legend is difficult to read.

Model Implementation Methodology

  • This section contains parts that should be included in the "related work" section, are repeated from the previous section (e.g. Agisoft Metashape) or are completely unnecessary because they were not practically used, e.g.:
    • Rows 252-258: move to the "related work" chapter or delete.
    • Rows 286-292: redundant text.
  • A detailed description of how the semantic data were created is missing.
    It is not clear to me in which stage of the process semantic data was created. Especially when SketchUp itself, as well as Autodesk 3ds Max, do not support semantics editing.

High-performance Visualization

  • A comment similar to the first one in the previous section applies:
    • Rows 399-413: Redundant text.
    • Rows 429-432: Redundant text.
    • Rows 472-476: Was this ADE used? If not, it is redundant information.
    • Rows 530-540: In my opinion, also a redundant text.
    • Rows 560-565: Too general text - redundant.
    • Rows 597-603: Was this compression used? If not, it is redundant information.
    • Rows 626-633: Redundant text.
  • Figure 21: This is the created LoD 3 model? This should be mentioned in the figure caption.
  • I miss any attempt to evaluate the created 3D models and visualizations.

Discussion

  • I would expect a discussion of the limitations (weaknesses, problems, ...) of the used procedures and the achieved results.

Conclusions and Future Work

  • Rows 717-718: Is interior (LoD 4) modelling relevant future work for an article in Remote Sensing journal?

Author Response

Dear Reviewer,

First of all, we would like to thank for your valuable contributions to our manuscript which increased its quality. We have modified the paper according to your suggestions as well as the inputs of other reviewers. Please also see our responses to your comments in the attached file. We would be grateful if you could review the modified version.

Kind regards,

Authors

Reviewer 2 Report

The work proposes the 3D modelling of a city composed by an existing part and a designed one, with the aim of visualization. A web platform and a gaming engine are tested for this purpose.

The motivation of the study, with the main objective being the visualization should be even clearer from the introduction, possibly explaining in higher detail how it could support the cited smart city functionalities.

In this sense I also suggest to change the title, especially removing the part about semantic data, which are kept, but treated and used very marginally in the study.

Several interesting topics are touched by the study, such as the need for integration and harmonization of two kinds of modelling, with different features (existing surveyed city objects vs designed city obejects); use of standardised data models to support interoperability (like CityGML) and related issues. Especially those two ones would deserve a wider part of the paper and the discussion.

A list of more detailed comments follow:

Line 19 - "future city design ... sense of place." This is a citation from the project within which the study was performed, I suppose. However it is not understandable from here, and this makes it meaningless for the abstract.

Line 49 - I suppose that here, with "models", "standardised data models" is intended. This is not clear in the text, the sentence should be rephrased.

Line 50-51 - The terms "simplified" and "personalized" are inaccurate, they should probably be "generalised" and "extended", with other references.

Line 51-52 - "new 3D city models..." The previous  versions of CityGML, as well as other data models for 3D city modelling, could allow this already. CityJSON introduces a more effective way of managing CityGML models, mainly by avoiding the use of GML, which could be tricky for the implementation. On the other hand, CityGML v.3 introduces additional features. These aspects should be explained better, if relevant. In any case, the sentence has to be corrected.

Line 61 - "to give a more realistic impression". Actually, city objects are represented if and when they have their specific role in the desired application or analysis (possibly being visualization).

Line 66 - "year of reconstruction, energy consumption, etc." are only about the semantics, you could add some example of topological information too.

Line 73 - "for stakeholders, decision makers and citizens" I would add "and for machines".

Line 76-77 - This sentence needs to be specified better, with a deeper insight and discrimination between what kind of models are actually available and what potential functionalities are shown at commercial events.

Line 83 - "smart buildings, smart energy, smart transportation" a few more details about the meaning of these terms would be required.

Line 92-93 "Only few examples..." - Actually, although being a relatively recent topic, a huge literature begins to be available. You can look for "BIM-GIS integration", "GeoBIM" and similar keywords.

Line 93-94 level of detail - It is necessary to say that this is defined according to the CityGML definitions (with reference).

Line 113 "has been limited to urban planning and visualization" - This is not true, just as an example, you can see the review of Biljecki et al, 2015, in your reference list already. Moreover, other applications of 3D city models exist (noise, energy, cultural heritage, etc).

Lines 116-120 - I think that here there is an overlap between the GIS concept as data (3D city models used for analysis) and GIS as the software currently available off-the-shelf. This difference should be clarified.

Lines 121-122 - You could also refer to the issues poited out to the GeoBIM benchmark project, about this point (https://3d.bk.tudelft.nl/projects/geobim-benchmark/presentations-publications.html) 

Line 129 and reference 32 - I don't think this claim is true or easily defensible. Moreover, the reference is not demonstrating this, maybe it can be moved somewhere else in the text.

Line 146 - These are not comparable, such as alternative platforms. The sentence should be rephrased.

Line 158 - "4D" should be shortly defined.

Line 164 and following - I don't think all these examples are based only on visualization of 3D city models.

Section 3 - Although Figure 2 is well designed and offers a meaningful overview of the methodology, the methodology steps have to be explained in higher detail, better if separating the different stages (subsections could be useful) and adding information about the methods, processing and choices made during the study, further than just citing the used software.

Line 196 - What does "in terms of geometric units" means in this sentence? It is not very clear.

Line 238 - Why "only"? What do you mean?

Line 239 - What are the differences between city plan and city model, in your sentence?

Line 268 - Who provided the building footprints? with which features? Metadata have to be explained and discussed. Same for the semantic data cited at line 271.

Sentence line 295-297 - Some reference would be helpful.

Line 360 "3D building models were placed manually" - in what environment or software was this done? Was any setting or specific processing necessary?

Line 381 "architectural models should be produced with the aim of real-time visualization" - What does it means concretely?

Line 400 - I'm not sure 57 is a proper reference for this sentence.

Line 402 "which demonstrates that the visualization is an important focus and a major value of such models" - I do not agree with this conclusion. Visualization is useful, but the fact that it's the most used application does not reflect the fact that it is the most important. For example, it is also the least challenging with respect to other kinds of analysis. The importance of visualization should be stated in a different way. Moreover, this is a relevant part about the motivation of this work, it is therefore supposed to be in the introduction.

Line 406 - does "level of detail" here mean the same than the CityGML LoDs? I think it's a different concepts and should be better explained.

Lines 441-442 - Can you discuss why did you need to have your data in CityGML if in the end you are converting them to a different format (3D tiles) for the use that you aim to (visualization)? It would be interesting.

Line 500 and Figure 13 - the height difference visible in the "unmerged" case should be discussed. What processing is applied by the Cesium ION platform? What happens?

Line 524 - The differences between ordinary and true-ortophotos should be introduced.

Line 579 - "atlassed textures"  have to be explained, and/or at least some reference explaining them must be added.

Line 614 and figure 18 - This is not a very relevant symbology to be applied, other examples could be more meaningful. Moreover, to show the effectiveness of your approach, integrating both the representation of existing buildings and the design of a planned district, it should be more effective to apply the same symbology to both parts of the city seamlessly.

Lines 640-641 - Some examples about the added value of this step should be explained.

Line 664 - How it can "facilitate participatory 664 planning and decision making"?

Line 679 and following - What do you mean with this sentence? Can you be more specific, please? I can't get the connection "SINCE CityGML.. , then, efficient database...". The sentence should be rephrased.

Line 684 - in what the "improved visual experience" consists?

Line 691 - the "requirements" here, are more the topics which are involved into the main research category "3D GIS".

Lines 696-698 - It is repeated in the discussion too.

Author Response

(The authors gave the same response as above.)

Reviewer 3 Report

It is a research article about reconstruction and visualization of heterogeneous (in terms of multiple level of details) 3D city models with semantic data. The English language and style of the manuscript is adequate to be a journal article. It is a well structured and described manuscript which gives the entire processing chain from data acquisition/processing/modeling to visualizing the city models. The state-of-the-art software and tools in industry and academia are linked to each other to establish the workflow. It is a complete study which may receive interest both from experts and non-experts working on 3D city modeling concept.

Some minor comments and recommendations are in the following.

- Title:
Some words in the title may be changed, such as;

Representation --> Visualization

Urban Scenes --> Urban Models
or
Urban Scenes --> City Models

- What is the difference between "semantic data" and "attribute data" in terms of GIS terminology? Is the term "semantic data" a deliberate usage in the entire text?

- 2. Background on 3D City Model Representations:

The procedural city/building modeling may be used not for the existing LoD2 city model but for the future smart LoD3 city concept. A short discussion with a few references might enrich the text.

* Procedural modeling of buildings. P Müller, P Wonka, S Haegler, A Ulmer, L Van Gool. ACM SIGGRAPH 2006 Papers, 614-623

** Procedural modeling of cities. YIH Parish, P Müller. Proceedings of the 28th annual conference on Computer graphics and interactive techniques, 2001.

- Lines 206-207:

When is the production date of the building footprints?
Are the acquisition date of the aerial photos and the building footprints conforming to each other?

- 4.1. Semi-Automatic model generation of existing 3D city model in LoD2:

736 UltraCam images were photogrammetrically processed using the Trimble Inpho software. Once the orientation information is extracted, the DSM was produced using the Agisoft Metashape software. This DSM was input to the lastools software to produce the DTM.
The public administration delivered building footprint files (ESRI Shapefile), the DSM and the DTM are fed to the BuildingReconstruction software to reconstruct the 1202 buildings of the test site.
This is the presented 3D building model generation methodology..
What is error assessment of this methodology? What are the Type-I and Type-II error rates. A confusion matrix or computation of the correctness, completeness and quality metrics may give useful information to gauge it.

-Lines 273-276:

Was there any reference frame difference between the building footprint Shapefiles and the DSM/DTM files, as they come from two different data sources?
If yes, how did you dissolve the problem?

-Lines 323-324:

The topology matters here.
Does the SketchUp software maintain a certain topology? Is there any better software than SketchUp in this respect?
Does the existing LoD2 city model has a correct topology?
Are the existing and the future city models topologically conforming to each other?

-Line 338:

with a a small number of polygons.
-->
with a small number of polygons.

-Lines 451-456:

A table or a list of bullets would be more styling than the plain text.

- 6. Discussion:

The Discussion chapter is too short, which lacks of necessary criticism and commentary. Especially the used software and tools should be classified based on their functions, and assessed based on their advantageous and disadvantageous points. The bullet lists or tables can be used for the presentation. They can be grouped into two, such as city model generation and visualization.

Author Response

(The authors gave the same response as above.)

Round 2

Reviewer 1 Report

Most of my suggestions have been responded in the text of the paper. The clarity of the article has increased. The paper is still relatively long, but now it is already within tolerable limits. I have only a few minor comments, that relate particularly to formal aspects of the paper.

Some abbreviations used in the text are not explained – i.e. UN, CityGML, UAV, LiDAR, X3D.

Figure 1 - the legend can also be placed outside the map field, then there will be no problem with readability.

Figure 7 - it is not entirely easy to compare the complexity of 3D models when each is viewed from a slightly different perspective.

Figure 21 – there is still missing the information, that the displayed model is LoD 3 mentioned in the text of the paper. Also the caption is practically the same as in the next figure (Fig. 22).

The Supplementary Materials section should be adjusted to the next phase of review.

Author Response

Dear Reviewer,

We would like to thank again for taking your time to review our manuscript for the second time and providing your contributions. We have modified the paper according to your suggestions. Please see our responses to your comments in the attached file. We would be grateful if you could review the modified version of the manuscript.

Kind regards,

Authors
